# Bionic Path Planning Fusing Episodic Memory Based on RatSLAM

**DOI:** 10.3390/biomimetics8010059

**Published:** 2023-02-01

**Authors:** Shumei Yu, Haidong Xu, Chong Wu, Xin Jiang, Rongchuan Sun, Lining Sun

**Affiliations:** 1School of Mechanical and Electrical Engineering, Soochow University, Suzhou 215137, China; 2School of Mechanical Engineering and Automation, Harbin Institute of Technology Shenzhen, Shenzhen 518055, China

**Keywords:** bionic path planning, RatSLAM, episodic cognitive map, connectivity networks

## Abstract

Inspired by rodents’ ability to navigate freely in a given space, bionavigation systems provide alternatives to traditional probabilistic solutions. This paper proposed a bionic path planning method based on RatSLAM to provide a novel viewpoint for robots to make a more flexible and intelligent navigation scheme. A neural network fusing historic episodic memory was proposed to improve the connectivity of the episodic cognitive map. It is biomimetically important to generate an episodic cognitive map and establish a one-to-one correspondence between the events generated by episodic memory and the visual template of RatSLAM. The episodic cognitive map can be improved by imitating the rodents’ behavior of memory fusion to produce better path planning results. The experimental results of different scenarios illustrate that the proposed method identified the connectivity between way points, optimized the result of path planning, and improved the flexibility of the system.

## 1. Introduction

With the continuous innovation and development of artificial intelligence technology, autonomous robots have appeared in agriculture, industry, medical, and military fields and are highly convenient for humans. Mapping, positioning, and path planning are key technologies in the field of mobile robots, and these are the current challenges in the research field [1]. The popular field of autonomous driving has increasingly highlighted the importance of path planning research. Existing results on path planning mainly involve road finding, choosing the safest strategy, and deciding on the most feasible trajectory [2]. Scholars have studied many traditional path planning algorithms, such as Dijkstra, A Star, Rapidly Exploring Random Trees, the Artificial Potential Field method, and Field D* [3,4,5,6,7]. However, compared with traditional path planning methods, research on bionic path planning is limited. Integrating biological characteristics into existing path planning algorithms is a feasible scheme to realize biological intelligence and improve its performance [8,9,10].

Currently, researchers have infused bionic characteristics into path planning algorithms. Experimental results show that it is reasonable to use a bionic concept to solve the path planning problem. Ji et al. proposed a hybrid bionic robot. The robot has lower limbs based on a chameleon and upper limbs based on a human as operating and travel mechanisms, which achieved desirable results [11]. Bangyal et al. proposed a new variant of BA to solve high-dimensional optimization problems by introducing a circle walk (TW-BA), which is superior to traditional bionic meta-heuristic algorithms. Population initialization is an important factor in meta-heuristic algorithms, and initialization based on low-difference sequences causes meta-heuristic algorithms to function better in global optimization problems [12,13]. Chen proposed an optimal path planning algorithm that combined the ant colony algorithm and genetic algorithm [14]. Xiang et al. applied a hybrid control method to a biped walking robot and raised a spatial planning structure, which can realize feedback control and other actions such as the motion of the robot’s legs to achieve highly efficient walking [15]. ANNs (artificial neural networks) are also used in biomimetic path planning to simulate advanced animal brains. On the basis of establishing an ANN, Dubey et al. solved the collision-free shortest path problem using the nonlinear function approximation method [16]. They applied this method to the manipulator of a mobile robot and obtained better results than the Q learning method. Qi et al. proposed the use of the ant colony theory to optimize the path planning algorithm. The effectiveness of the algorithm was tested in dynamic environments [17]. Pshikhopov introduced some bionic path planning methods, including neural networks, neural structures, and genetic searches, in their review paper on vehicle path planning and provided appropriate data tools for researchers to study bionic path planning [18]. Although there are many methods for solving the problems faced by bionic path planning, there is little research on bionic path planning combined with episodic memory. Existing solutions to challenges in bionic path planning have not yet been used to form a holistic bionic system. This paper provides a solution to the path planning problem by integrating episodic memory and SLAM modules from the perspective of bio-navigation.

Episodic memory can provide an accurate and simple method for path planning from the perspective of bionics. Episodic memory, first identified by Turing and Endel in 1972, is a declarative or explicit memory [19]. Since episodic memory is an indispensable part of the human memory system, we must integrate knowledge from past experiences and flexibly use it to achieve our goals in order to survive. The experience information stored in episodic memory mainly reflects past events and situations in time and space and is the most advanced and mature memory system in human beings [20,21,22]. The generation of episodic memory is closely related to the hippocampus and the entorhinal cortex of mammals. It can effectively control behavior through memory and reorganization events. On the basis of Turing’s research on the theory of episodic memory [23,24], other scholars have expanded the research related to episodic memory from different aspects; for details, refer to [25,26,27,28]. Creating robots that think and store memories in the same way that human beings do is a challenge that researchers are keen to address. Episodic memory stores the impression of a scene in an organism, and the organism looks for a path by using these impressions; therefore, episodic memory has a guiding effect on the path planning of an organism. Without episodic memory, there would be no context for the memories created by an organism, and there would be no way of avoiding repetition. Although episodic memory provides a favorable method for path planning, it also requires a bionic mapping system as a carrier to form an overall bionic path planning system. RatSLAM, a bionic navigation system with synchronous localization and mapping whose output is a precise topological map, is a reasonable carrier for applying the theory of episodic memory and building an episodic cognitive map.

Constituting a popular research topic over the past 10 years, the robot navigation system with bionic characteristics known as RatSLAM was the result of a collaboration between artificial intelligence and the field of biological nerves, and has huge potential in future research. The RatSLAM system is a computer model that simulates the hippocampus of rodents [29]. In the system, mapping from pose cells, head direction cells, and grid cells in the hippocampus of rodents to the computer model was established to realize bionic drawing [30]. The RatSLAM system generated visual information through the camera, calculated the visual odometer as the input, and used the SAD (sum of absolute difference) visual template matching method and loop-closure detection to produce a more accurate experience map with location information [31]. To enhance the robot path planning method and increase algorithm flexibility, we aim to integrate the memory and incorporate the functions of human beings into the navigation systems of bionic robots. Therefore, it is worthwhile to explore the combination of the RatSLAM system and episodic memory. It is valuable to employ unique bionic technology to process the experience map built by the system imitated by the rat hippocampus and to connect with the episodic memory model related to biological characteristics to produce a kind of cross-domain integrated intelligent system.

Compared with traditional SLAM solutions, RatSLAM uses a three-dimensional cell structure with biological characteristics rather than Cartesian grids; therefore, it can produce environmental models with more accurate topology information. Since the RatSLAM system generates simple and accurate topological maps, it is appropriate and feasible to combine the RatSLAM system with episodic memory as a way to improve the bionic path planning algorithm. This paper proposes a combination of path planning, episodic memory, and RatSLAM to form an overall bionic path planning system. A connectivity network is also proposed to improve the accuracy of path planning.

In this paper, a bionic path planning solution that uses an episodic cognition map based on the RatSLAM system is provided. A connectivity network model fusing historic episodic memory to enhance the accuracy and intelligence of the bionic path planning algorithm is also proposed.

The remainder of this paper is arranged as follows. Section 2 describes the bionic path planning method based on episodic memory. Section 3 introduces an improved bionic path planning method fusing historic episodic memory. Section 4 presents experimental results to verify the proposed method. Finally, Section 5 summarizes our study.

## 2. Bionic Path Planning Based on Episodic Memory

Taking advantage of the ability of episodic memory to connect past and future events, path planning incorporating bionic functions makes it possible to achieve better results by building a connectivity network.

### 2.1. Concept of Episodic Cognitive Map

The episodic cognitive map is a spatial representation method describing the environmental spatial information and the correlation of adjacent scenes [32]. It has the same structure as the topological map produced by RatSLAM that consists of a number of experience points indicating the robot’s posture and visual templates representing how the robot observes the environment. Therefore, an episodic event can be established as *ε_i_
*= {*O_i_*, *S_i_*, *P_i_*}, where *O_i_* is the visual template in RatSLAM, *S_i_* represents the activity of the *i*th episodic neuron, and *P_i_* contains the robot’s posture [33,34,35]. An episodic cognitive map is a two-dimensional matrix describing the weights of connections between episodic neurons, as expressed in (1). In our study, the diagonals of the cognitive map *M* represent the activity of different episodic neurons that are most active at birth and then decline over time. When the robot arrived at a previously visited location, the episodic neurons generated at that location were reactivated at maximum activity.
(1)$$M=\left[\begin{array}{ccccccc} S_{1} & W_{12} & W_{13} & \cdots & W_{1i} & \cdots & W_{1m}\\ W_{21} & S_{2} & W_{23} & & W_{2i} & & W_{2m}\\ W_{31} & W_{32} & S_{3} & & W_{3i} & & W_{3m}\\ \vdots & & & \ddots & & & \vdots \\ W_{i1} & W_{i2} & W_{i3} & & S_{i} & & W_{im}\\ \vdots & & & & & \ddots & \vdots \\ W_{m1} & W_{m2} & W_{m3} & \cdots & W_{mi} & \cdots & S_{m} \end{array}\right]$$

Episodic neurons *S_i_*(*t*) are most active at birth and then decline over time. *W_ij_*(*t*) represents the connection weight between episodic neurons: *S_i_*(*t*) and *S_j_*(*t*). It implies a correlation between the positions of robots that generate both episodic neurons. The change in activity of the episodic neurons is expressed in (2):(2)$$S_{i}\left(t\right)=\left\{\begin{array}{c} DS_{i}\left(t-1\right)\;\\ 0 \end{array}\begin{array}{c} \left(S_{i}\left(t-1\right)>\theta \right)\\ \left(S_{i}\left(t-1\right)\leq \theta \right) \end{array}\right.$$

Here, *θ*, representing the minimum value of activity, is the memory depth of the episodic neuron. The attenuation coefficient *D* represents the attenuation weight of the episodic neuron, and τ represents the attenuation constant:(3)$$D=e^{-\tau }$$

### 2.2. Information Expansion of Episodic Cognition Map

The episodic cognition map established for simple environmental path planning is not perfect, and the only connection weight is found between path points. We simply integrated the distance information and the episodic cognition map and expanded the original episodic cognition map on the dimension so that it contained more information related to path points. In an episodic event, the episodic neuron corresponds to the posture of a robot in the experience map. The second dimension of the extension is the Euclidean distance between the corresponding experience points of each episodic neuron in the experience map. Therefore, the extended episodic cognition map contains not only the correlation information of connection weights of the path points but also the distance information between the road points. The second-dimension episodic cognitive map *M_2_* is shown as (4):(4)$$M_{2}=\left[\begin{array}{ccccccc} 0 & L_{12} & L_{13} & \cdots & L_{1i} & \cdots & L_{1m}\\ L_{21} & 0 & L_{23} & & L_{2i} & & L_{2m}\\ L_{31} & L_{32} & 0 & & L_{3i} & & L_{3m}\\ \vdots & & & \ddots & & & \vdots \\ L_{i1} & L_{i2} & L_{i3} & & 0 & & L_{im}\\ \vdots & & & & & \ddots & \vdots \\ L_{m1} & L_{m2} & L_{m3} & \cdots & L_{mi} & \cdots & 0 \end{array}\right]$$
where *L_ij_* and *L_ji_* are equal, and they are the Euclidean distances between the experience points corresponding to *S_i_* and *S_j_* in (1). *L_ij_
*is expressed by (5) as follows:(5)$$L_{ij}=\sqrt{{\left(e_{i}\left(x\right)-e_{j}\left(x\right)\right)}^{2}+{\left(e_{i}\left(y\right)-e_{j}\left(y\right)\right)}^{2}}$$
where *e_i_*(*x*) and *e_i_*(*y*) represent the locations of *e* on the two axes in the experience map.

### 2.3. Path Planning Using Episodic Cognitive Map

An episodic cognitive map made by simulating an episodic memory function can be applied to robot path planning with a topological structure. Episodic neurons in the episodic cognitive map correspond one-to-one with the nodes in the experience map, which is similar in every instance that a robot observes scene. For every experience point generated in RatSLAM, an episodic neuron will appear in the episodic cognitive map corresponding to it. Episodic events not only generate the environment map of the current scene but also generate memory storage for the images seen by the robot, and then fuse the stored memory with location information to generate the expanded episodic cognitive map. An episodic cognitive map reflects the connection of spatial and temporal experiences, facilitating the planning of optimal paths.

A navigation flow chart based on situational memory is shown in Figure 1. When a navigation task is assigned, the target-related memory is extracted from the episodic cognition map, and a better path is proposed based on the activation sequence of neurons. First, according to the current scene and the target scene, the corresponding initial neuron and target neuron are located in the M matrix. Then, the neuron with the highest connection strength to the initial neuron is selected as the next activated neuron. If there are multiple neurons in a row and the connection weight of the current neuron is the maximum weight, the neuron with the smallest distance from the target neuron according to the neuron serial number is located. Finally, this process is repeated to find the neuron with the largest connection weight and the smallest distance until the target neuron is found.

Path selection was carried out according to the sequence of experience points or the distance from the end in the past. While it is easy and quick to use experience point serial number connections for backtracking, this method can produce errors when paths are repeated in different directions. On the other hand, when confronted with a complex fork in the road, choosing the road point based on the experience point connection using only the distance from the end point can create a directional error. Therefore, those methods cannot complete the task in a complex environment where the road continues. To solve the problem of a complex environment and sequence, we built a connectivity network to increase the number of alternative roads and use different probabilities to choose the corresponding road.

## 3. Path Planning Fusing Episodic Memory

### 3.1. The Architecture of Connectivity-Improved Episodic Cognitive Mapping

#### 3.1.1. The Idea of Connectivity Networks

The connectivity network is a two-dimensional computing network model based on the characteristics of neurons used to determine the possibility of connectivity between path points. By fusing episodic memory, a connectivity-improved network is established to increase the level of environmental information in the episodic cognitive map.

Two intersections can be identified when a person stands at both ends of an intersection and watches without crossing the road that connects the intersections. Thus, if a robot has the same ability to recognize intersections as human beings, it can plan the optimal path without crossing the entire road. This will greatly improve the efficiency and accuracy of path planning. The blue line shown in Figure 2 represents the path traveled by the robot. It starts from starting point A, travels in the direction of the arrow, and ends at point A to form a loop closure. Our goal is to plan an optimal path from starting point A to end point B based on the blue topological path. Only through the combination of the RatSLAM system and episodic memory can the red dotted line A→D→B be planned. However, in the map, some roads are not passed by the robot, and the existence of these roads will affect the final path planning results, such as the black dotted line C→B shown in Figure 2. Therefore, to improve the accuracy and intelligence of path planning, we used a connectivity network of roads so that the robot does not cross the road but waits and observes an action at points C and B at both ends of the road. Under the circumstances that the robot achieves the above task and the matching algorithm is applied, the system can judge whether there is a connected road between C and B. The algebraic results of the connectivity network are used to express the connection weights at both ends of the road. The larger the result value, the higher the activity of connected neurons in the connectivity network, and the greater the possibility of roads between the two points. After establishing the connectivity network and updating the episodic cognitive map through the connected neurons, the optimal path of the black dotted line A→C→B can be planned in the environment shown in Figure 2.

In order to improve the accuracy and flexibility of path planning, this paper integrates episodic memory, associative memory, and a neuron model to establish a connectivity network in order to evaluate the correlation between road points in the trajectory, and use this to update the episodic cognitive map. In the proposed method, when a robot has seen a road at an intersection, it will save the information of that intersection to form a “memory”. On the other hand, when a robot has reached the other end of the intersection and looked back at the road but does not cross it, a robot can recognize that the road has existed in the past. The proposed connectivity network allows robots to plan paths more intelligently and flexibly by enabling them to find potential paths without having to travel those paths in advance.

#### 3.1.2. Introduction to the Connectivity Network

The connectivity network is a bionic network model that combines episodic memory, associative memory, and neurons. Associative memory is a type of memory and recall in which people recall memories via a background, context, or certain clues [36]. Similarly, when a robot with a connectivity network crosses the point that closes a loop again, it will recall the relevant clues and revive the memory of the current scene. In this process, there is a connection between the two “experience points”. Once this kind of associative memory occurs, there is usually a connection between the two experience points, which can even be connected by a potential path that has not been traveled before. Brains are able to recall specific events in specific places and times and make connections between these experience points [37]. From the perspective of episodic memory, the connectivity network aims to establish a connectivity between two events in the network. Episodic memory provides a seemingly infinite storage space for daily experiences and a retrieval system so that accessing these experiences while partially activating them is possible [21]. The connectivity network functions by using such a retrieval system to access these experiences and judge their similarity. There is connectivity between nervous systems, and nerve cells are associated with memory [36]. Connectivity networks are made up of such neurons with memory associations.

After matching the visual template in the system, connectivity networks can match two specific episodic memory events together and establish connectivity. If the highest connected neuron activity in the connectivity network reaches a threshold value, a potential path may exist between the two corresponding episodic memory events. 

Figure 3 depicts the architecture of a connectivity-improved mapping system that integrates the RatSLAM system and episodic memory. Pose cells with a three-dimensional structure generate an experience map through the calculation of visual odometer and path integration. Meanwhile, the input image is built into a local visual cell sequence to generate a visual template for storing environmental information. Inspired by episodic memory, episodic events containing episodic information are constructed by visual templates in the RatSLAM system. An episodic cognitive map is composed of episodic neurons, which reflect the weight of the connection between various episodic events. In other words, an episodic cognitive map is a concept that describes the correlations between episodes. The connections between the experience points in the experience map constitute the connected neurons in the connectivity network, and the episodic cognitive map is updated when the connected neurons break the threshold and are confirmed to be connected. Connections between connected points are established in the updated episodic cognitive map, which determines the connectivity of the road without crossing it.

After the connectivity network is added to the system, operations such as visual template matching, threshold selection, and connectivity network excitation and suppression are carried out, and the experience points corresponding to episodic neurons that break through the connectivity threshold are connected.

In the episodic cognitive map, spatial connections can be established between two episodic neurons. Furthermore, the connections between their neighboring neurons should be the same to maintain spatial connections. The proposed method updates the episodic cognitive map so that the untraveled but connected roads in the environment also have real connectivity weights, increasing the flexibility and accuracy of path planning. Humans can recognize an untraveled road by comparing both sides of it. This inspires us to use visual template matching on both sides of a potential road to determine its connectivity. In this case, because of observing the road from an opposite perspective, one of the visual templates should be flipped when templates are being matched. The connectivity network considers the results of SAD (sum of absolute differences) matching.

### 3.2. Neural Stimulation of Memory Connection Networks

The connectivity network is composed of connected neurons, which conform to the excitation characteristics of neurons. As opposed to episodic neurons, connected neurons exist in connectivity networks, and their activity depends on the probability of the connectivity between the corresponding experience points. They are used to represent the connectivity between road points in a space. Figure 4 shows the change in membrane potential over time in a single-neuron model after applying excitation. The neuron receives the past output as the input. Its membrane potential suddenly changes and generates the action potential transmitted along the axon when the accumulated value of the input increases to a threshold. This is also known as the pulse [38]. A connectivity network is composed of several single-neuron models named connected neurons. Each connected neuron corresponds to two experience points, and the activity of the neurons is proportional to the probability of connectivity between experience points. When a robot generates episodic events to record its current location and visual template, an episodic neuron is generated. At the same time, a connected neuron in the connectivity network is also generated to describe the probability of connectivity to other experience points. Each neuron sends an excitation signal to the other neurons. This excitation correlates with the two-dimensional Gaussian distribution shown in (6):(6)$$J\left(x,y\right)=A_{m}e^{-\left(\frac{{\left(x-x_{0}\right)}^{2}}{2\sigma_{x}^{2}}+\frac{{\left(y-y_{0}\right)}^{2}}{2\sigma_{y}^{2}}\right)}$$
where *J*(*x,y*) represents the excitation generated by the connected neuron at position (*x_0_, y_0_*) to the connected neuron at position (*x,y*), *A_m_* represents the amplitude of the excitation, and *σ_x_* and *σ_y_* represent the standard deviation. Each connected neuron generates a certain amount of excitation to the surrounding neurons. The generated activity of the connected neuron remains unchanged, but the excitation applied by the neuron decays over time, similar to the attenuation in the single-neuron model. The activity of a connected neuron is denoted by (7).
(7)$$A_{ij}\left(t\right)=A_{ij0}+S_{w}\left(t-t_{0}\right)J_{0}e^{-n\left(t-t_{0}\right)}+\cdots +S_{w}\left(t-t_{q}\right)J_{q}e^{-n\left(t-t_{q}\right)}$$
where *A_ij_*(*t*) is the activity between the *i*th and *j*th connected neuron at time *t, A_ij0_* is the initial activity, and *J_q_* is the first *q* incentive, while *n* controls the attenuation speed. *t_0_* is the moment of the first excitation. *Sw*(*t*) is the switching function represented by (8), which controls whether excitation is applied:(8)$$S_{w}\left(t\right)=\left\{\begin{array}{c} 1\;,\\ 0\;, \end{array}\right.\begin{array}{c} t>0\\ t\leq 0 \end{array}$$

In addition, each generated neuronal excitation has a neuronal decay feature to ensure that the activity of the connected neurons is not infinite. The connection of connected neuron activity was used to determine whether the corresponding waypoints connected.

As shown in Figure 5, each circle in the connectivity network represents a connected neuron whose active value is represented by its depth of color. Each connected neuron in the network corresponds to the connection between two experience points that contain road information. The connected neurons with larger weights in the connectivity network have stronger connections between experience points. This is also shown in the thicker lines in Figure 5.

If the result of mirror matching exceeds a given threshold in advance, the two locations corresponding to the connected neuron are regarded as connected; a potential path between them exists.

### 3.3. The Probabilistic Acceptance Model

Inspired by the biological phenomenon that humans can recognize the same road when they can see both ends of it, visual template matching is used at both ends of a road to determine the connection between these two points. Every time a robot collects data and builds a map in a large environment, the number of visual templates that it builds increases with time. However, matching these visual templates one-by-one greatly reduces the efficiency of path planning. Therefore, it is necessary to reduce the matching range of the visual template to speed up the proposed method. Road join points are located at the intersection, and thus, from the perspective of bionics, the range of visual templates are matched by capturing the action of the robot looking at the intersection, in the same way that a human would.

As shown in Figure 6, the system establishes a path from experience point 1 to experience point 17 in the direction of the black arrow, and experience point 17 is connected to experience point 4. Our goal is to plan an optimal path from blue number 1 to orange number 15. Experience point 4 is connected to experience points 3, 5, and 17, and selecting the next feasible point based only on the principle of closest distance from the starting point will lead to the incorrect choice of experience point 5. This contradicts the actual optimal path represented by the red dotted line. Increasing the probability in the model of road choice is a way to solve this problem.

A probabilistic selection model is established to give all points around the selection a certain probability, and the probability is negatively correlated with the distance from the destination. The probability *P_q_* of each connected experience point being selected is shown in (9):(9)$$P_{q}=\frac{I\left(q\right)}{I\left(1\right)+\cdots +I\left(q\right)+\cdots +I\left(n\right)}$$
where *n* represents the number of connected points, and *I*(*q*) is the probability index, whose value affects the probability of experience points *e_q_*. *I*(*q*) is represented by (10), where *L*(*q*) is the distance from the starting point:(10)$$I\left(q\right)=e^{-L\left(q\right)}$$

The closer the road is to the starting point, the more likely it is to be chosen. This relationship should not be linear, so that the exponent is suitable for expressing the relationship. Since the choice of road points is probabilistic, it is necessary for the system to carry out multiple iterations, and the shortest path can be planned by comparing the total length of road after the iteration. In addition, to avoid the infinite loop caused by going back and jumping out of the local optimal problem, every time the next point in the episodic cognitive map is determined by the connection weight, the connection is cleared to zero. The pseudocode of the algorithm is shown in Table 1.

### 3.4. Analysis of the Proposed Algorithm

The integration of the connectivity network and episodic cognitive map enables road points in order to determine the temporal order caused by episodic memory and establish the spatial connection caused by connectivity. The addition of connectivity improves the connection network of road points in the environment. The establishment of the connected neuron model is the realization of a computer model with a bionic mechanism, which reproduces the excitation mechanism of neurons, increases their activity through the excitation generated by membrane potential signals, and determines whether there is connectivity between roads by comparing the activity with the set threshold value. Not only are there connections between road points and their surroundings, but the connected points also build a “bridge” for a long distance. Therefore, RatSLAM, episodic memory, and connectivity networks are closely connected and cannot be separated from one another. The robot can map out the road without crossing any part of it, which increases the flexibility and intelligent characteristics of path planning. The combination of RatSLAM, episodic memory, and connectivity networks effectively improves the whole bionic system, improves the accuracy of path planning in a complex environment, and enhances the robustness of the algorithm. The application of the connectivity network in the RatSLAM system enables the robot to recognize the existence of the road without crossing it, which further promotes the intelligence of the RatSLAM system and contributes to the improvement of robot path planning.

## 4. Experiment with Connectivity Network

The function of the connectivity network, which is to update the connection weight of episodic cognitive maps, and the improvement of path planning through the method of road selection based on probabilistic models are in urgent need of experimental verification.

### 4.1. Experiment Scene Collection

One set of outdoor data that we collected and one set of open data collected in St Lucia, Brisbane, Australia were used to assess the feasibility and accuracy of the proposed algorithm. An Osmo Pocket camera (Shenzhen, Dajiang Innovation Company, China) with a field of view (FOV) of 80°, a sampling frequency of 25 Hz, and a resolution of 1080 p were used to collect video data in our campus.


*First scene: The flower bed in front of the library*


As shown in Figure 7, a closed-loop road is formed with point A as the starting point and point B as the end point. The red line shows the path, and the arrow shows the direction. The blue dotted line is the path of the closed-loop part, the black dotted line between C and D is the path that has not been traveled, and there is a connection between them.


*Second scene: an abandoned driving school at Soochow University*


To further explore the superiority of the algorithm, the scale of the experimental scene was expanded, and a scene with a strong interference was selected, as shown in Figure 8. A and B represent the starting point and the end point, respectively. The solid red line represents the path of the first trip, the arrow represents the direction of the trip, and the dashed blue line is the repeated route generated after the closed loop. C and D are the two ends of the road with connectivity. The robot needs to turn and observe both sections of the road so that the system can calculate the connectivity of the road.


*Third scene: Part of the suburb of St Lucia, located in Brisbane, Australia*


A part of the dataset of St Lucia suburbs, collected at 3.45 pm, was chosen to verify the correctness of the algorithm in a more complex scene. The road map of the public dataset is shown in Figure 9 below, where point A is the starting point, and point B is the target point. The starting point and end point are selected in this dataset, and the path planning experiment is completed.

### 4.2. Analysis of Experimental Results

#### 4.2.1. Experiment A Verifies the Path Planning of the Probabilistic Mode

As shown in Figure 10, to plan the optimal path from A to B in the first scene, the path planning method based on episodic memory before and after improvement is selected in the system. Point C in Figure 10 is the intersection point, and the path in red is the result of a path planning algorithm. The results presented in the two pictures are inconsistent because the path planning method of episodic memory adopted in Figure 10a is selected based on the distance from the starting point, which causes the problem described in Figure 6, and the problem of contradiction in the sequence of events cannot be avoided. Figure 10b introduces the probability acceptance model, making it possible for point C to choose all connections when choosing adjacent connections. After several iterations, the shortest path can be calculated. Table 2 shows that the planned path of the proposed method has fewer turns and a shorter distance, and, thus, is evidently superior to the previous path.

#### 4.2.2. Experiment B That Verifies the Connectivity Network

Figure 11a is the output result of the visual odometer, and the yellow part is the corresponding segment of the intersection with watching action identified by the algorithm. Figure 11b is the experience map of the corresponding experimental scene output by the system. Figure 11c shows the path from starting point A to end point B planned by path planning and based on episodic memory, as proposed by [39]. Figure 11d shows the optimal path planned by the system after integrating the connectivity network. When comparing Figure 11c,d, it can be found that the two small pictures all arrive at the same end point from the same starting point. However, the path planned in Figure 11d shows the connected points in the path and planned the optimal path through the connections between the connected point C and point D. Table 3 shows that a path planning method-integrating connectivity network can provide a path with shorter distance and less turns.

Figure 12a,b show the visual template pixel pictures generated by the robot looking at both ends of the road, respectively. A road was observed in the image frame where the connecting points obtained by the system algorithm are located.

To verify the robustness and environmental adaptability of the algorithm, the second scene, where the road image was not as clear as the first scene, was chosen to test the feasibility of the algorithm. The experiment was carried out under the second scene, and four different kinds of outputs are shown in Figure 12. Figure 13a is the output of the visual odometer, the yellow part is the segment with road connectivity, and Figure 13b shows the experience map of the scene. Figure 13c shows the path planning algorithm proposed by [39] with a path that runs from point A to end point B, and Figure 13d shows the network connectivity of the optimal path. Figure 13c,d show that the system with the connectivity network can include the connected roads in the planning path, and a shorter optimal path can be planned. The path planning results of the second experimental scene are also shown in Table 3. By comparing the algorithms before and after improvements, it can be found that the method using the connectivity network yields better results.

Figure 14 is the image frame corresponding to the road connection points selected in the second scene. We observed the image and found that the system can also recognize the road connection in the scene where the road is slightly blurred to optimize the path planning.

In the third experimental scenario shown in Figure 15, point A and point B, which have a large distance span, were selected as the starting point and the end point, respectively. Through our improved path planning algorithm, the optimal path in red was obtained in the third complex experimental scenario. In Figure 15, the distance between A and C on the red line segment is 1926 m, the distance between A, D, and C on the blue line segment is 1983 m, and the distance between A, E, and C on the blue line segment is 2059 m. From the perspective of planning distance, this is the shortest path. Through the connectivity network model, a connection is established between points C and B in the episodic cognitive map to find a potential path, which is an unknown track in the cognitive map. The red line segment in Figure 15 shows that the path planned by the algorithm proposed in this paper has the shortest distance.

## 5. Conclusions

Guided by the episodic memory of biology, the connectivity network, which combines episodic events and connectivity to update the episodic cognitive map, is introduced on the basis of the RatSLAM bionic navigation algorithm to make the system plan a better path on the topological map. Firstly, the concept of connectivity networks is proposed, and a model of connectivity networks with neuronal excitation is established. Secondly, experienced maps and episodic cognitive maps are combined to expand the dimensions of episodic cognitive maps and increase the information content of episodic cognitive maps. In addition, a probabilistic acceptance model is used to improve the path planning algorithm. Finally, experiments on two scenarios verify the feasibility of the proposed algorithm, which indicates that the system is robust and highly scalable, and can successfully identify connected roads and complete the task of path planning in different environments. The navigation method based on the episodic cognitive map looks for a track from the starting point to the target point in the existing track. Episodic memory path planning based on a connectivity network may find a potential path that is unknown in the cognitive map, shortening the planned path distance. In future research, in order to conduct an experimental validation in a more complex environment, image segmentation and image recognition should be integrated in the algorithm. The robot can use this improved connectivity network to more easily establish the connection between roads, identify intersection information more intelligently, and plan a better path.

## Figures and Tables

**Figure 1 biomimetics-08-00059-f001:**
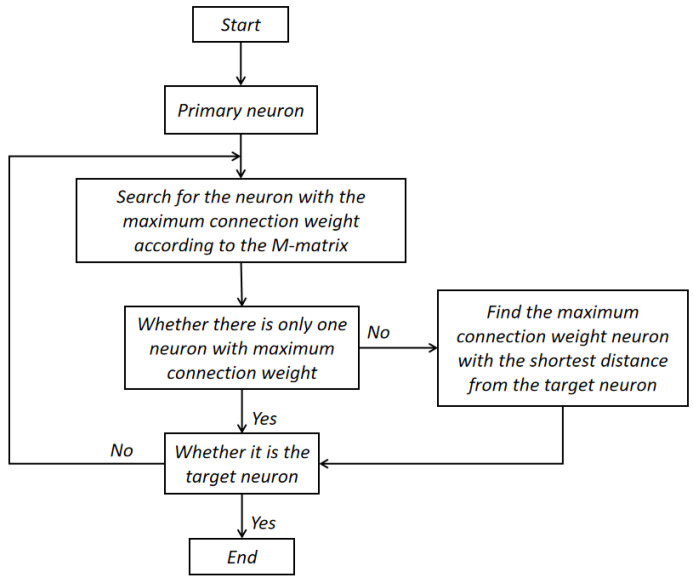
Flow chart of situational memory navigation.

**Figure 2 biomimetics-08-00059-f002:**
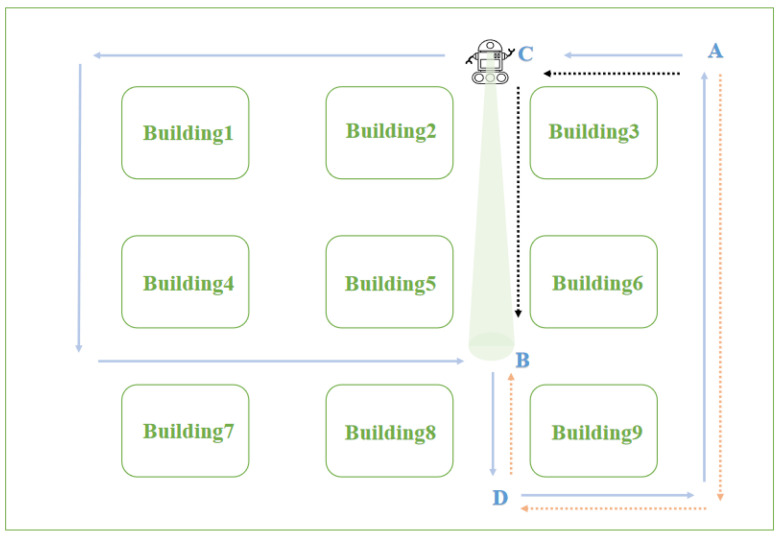
Schematic diagram of a robot system using a connectivity network.

**Figure 3 biomimetics-08-00059-f003:**
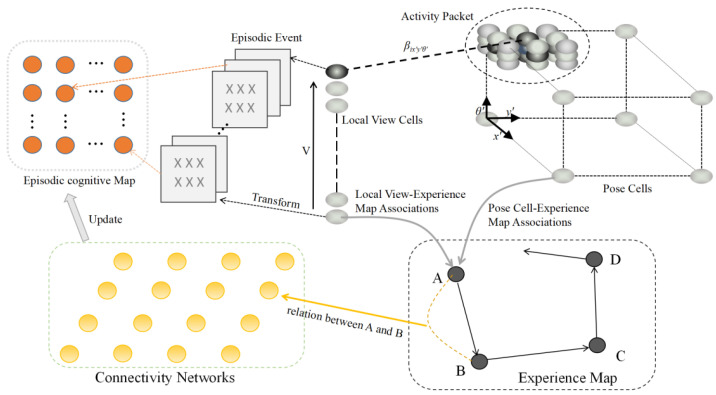
The architecture of the connectivity network, episodic memory and RatSLAM system.

**Figure 4 biomimetics-08-00059-f004:**
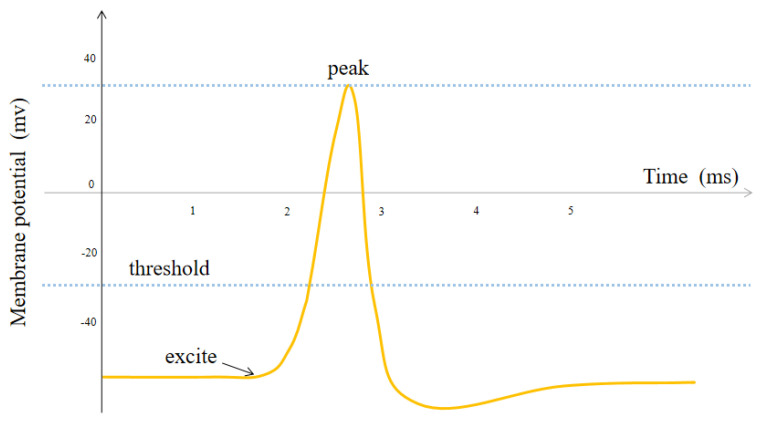
Single-neuron excitation model.

**Figure 5 biomimetics-08-00059-f005:**
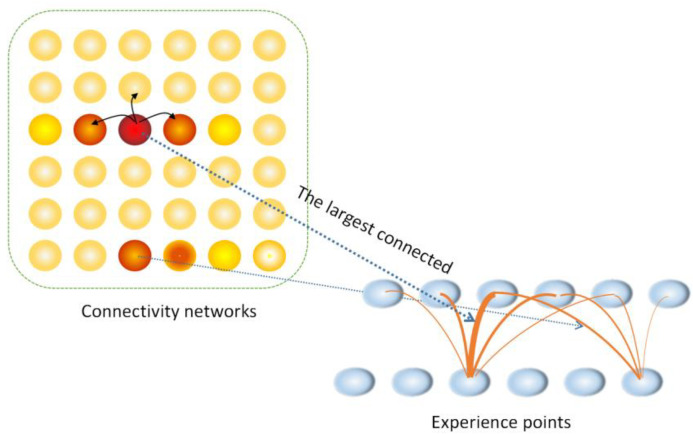
Schematic diagram of connectivity network.

**Figure 6 biomimetics-08-00059-f006:**
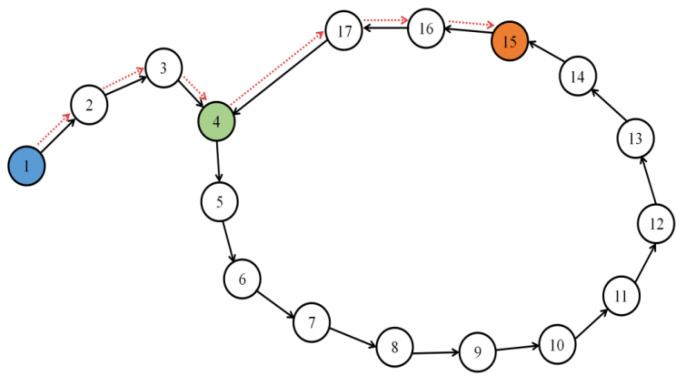
Path planning schematic of topological map.

**Figure 7 biomimetics-08-00059-f007:**
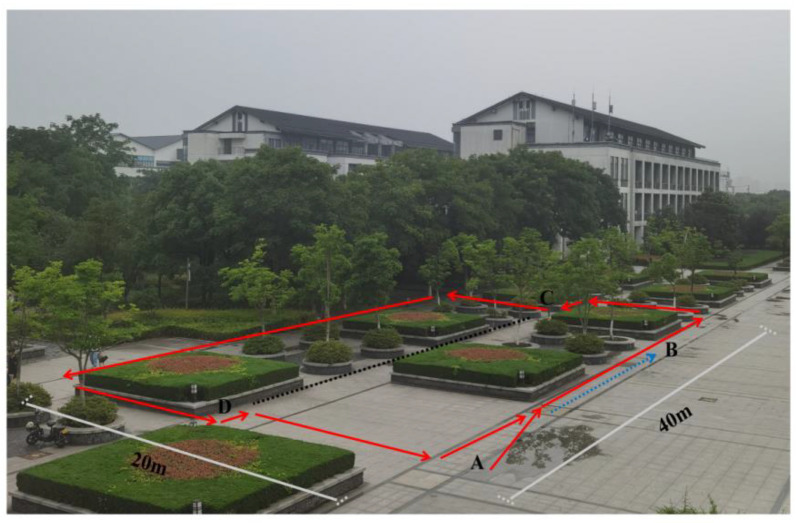
The flower bed in front of the library.

**Figure 8 biomimetics-08-00059-f008:**
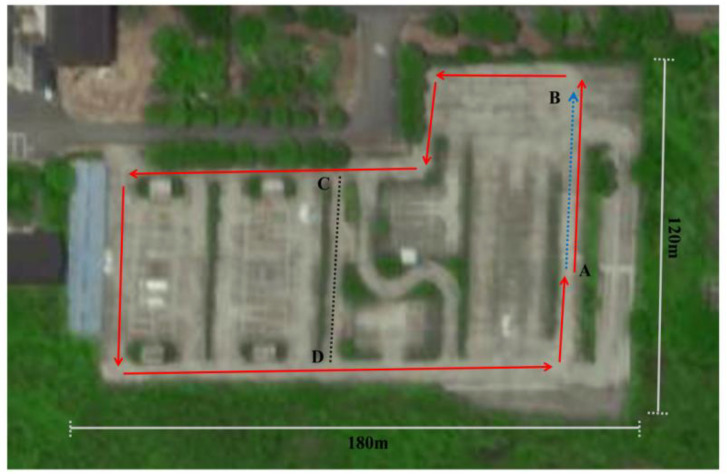
An abandoned driving school at Soochow University.

**Figure 9 biomimetics-08-00059-f009:**
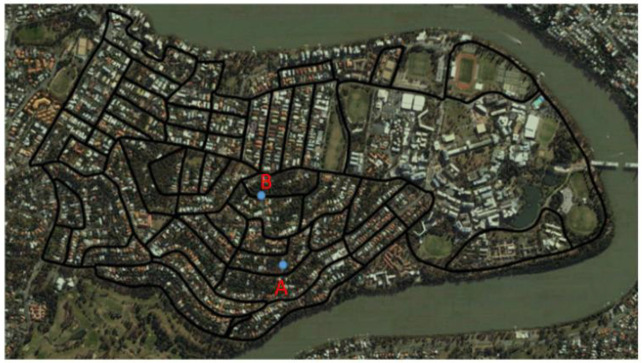
Saint Lucia open dataset roadmap.

**Figure 10 biomimetics-08-00059-f010:**
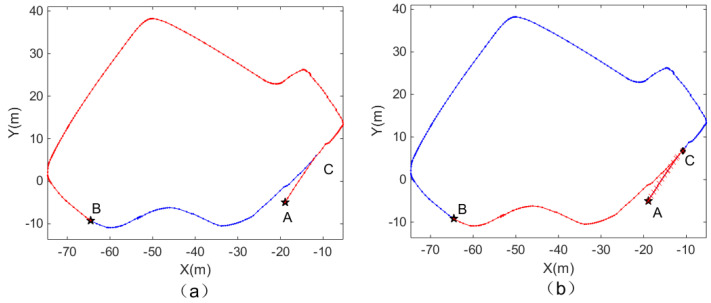
Path planning algorithms before and after model improvement. (**a**) Path planning based on episodic memory. (**b**) Path planning based on episodic memory with probability acceptance model introduced.

**Figure 11 biomimetics-08-00059-f011:**
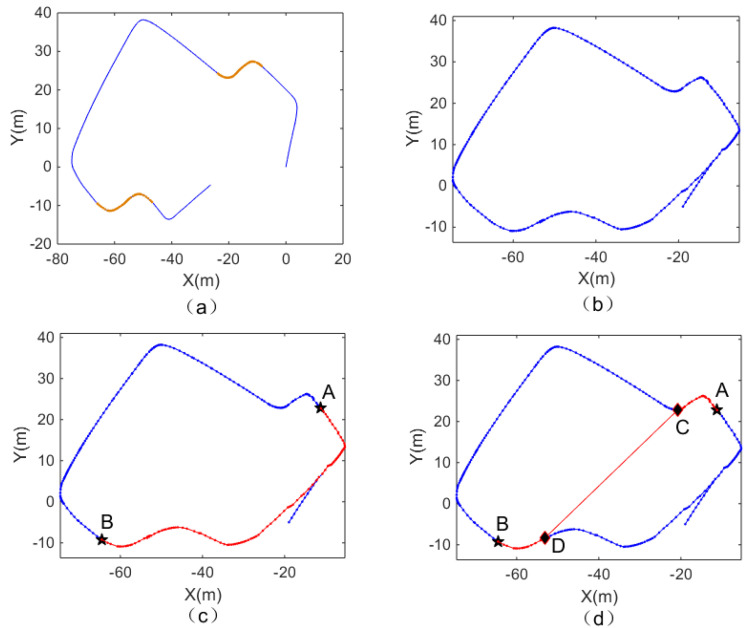
The output of the RatSLAM algorithm and the path planning results before and after the integration of the connectivity network of the first scene. (**a**) Output of the visual odometer. (**b**) Experience map of the first scene. (**c**) Path planning based on episodic memory. (**d**) The optimal path planned after introducing the connectivity network.

**Figure 12 biomimetics-08-00059-f012:**
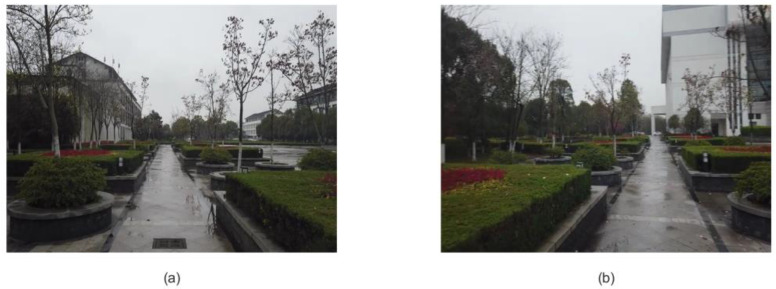
The image frame at the connected interface: (**a**) picture taken by the robot at one end of the road; (**b**) picture taken by the robot at the other end of the road.

**Figure 13 biomimetics-08-00059-f013:**
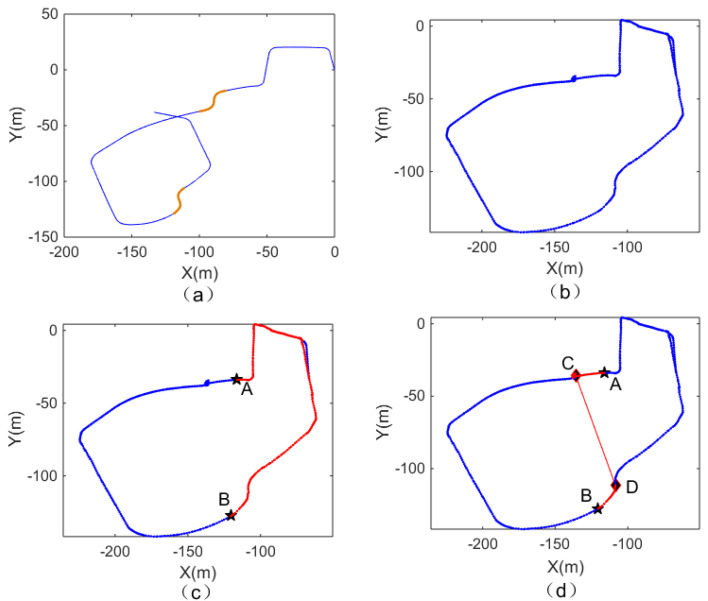
The output of the RatSLAM algorithm and the path planning results before and after the integration of the connectivity network of the second scene. (**a**) Output of the visual odometer. (**b**) Experience map of the first scene. (**c**) Path planning based on episodic memory. (**d**) The optimal path planned after introducing the connectivity network.

**Figure 14 biomimetics-08-00059-f014:**
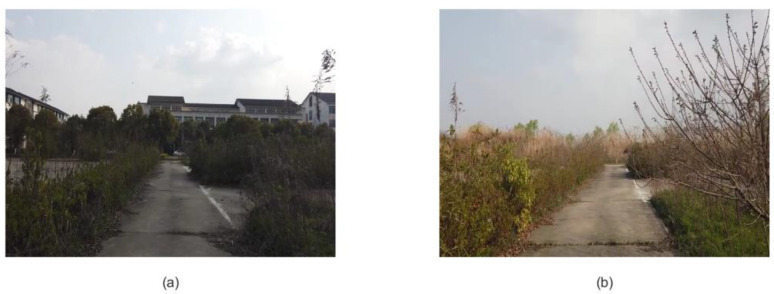
The image frame at the connected interface of the second scene. (**a**) Image frame obtained at one road connection. (**b**) Image frame obtained at another road connection.

**Figure 15 biomimetics-08-00059-f015:**
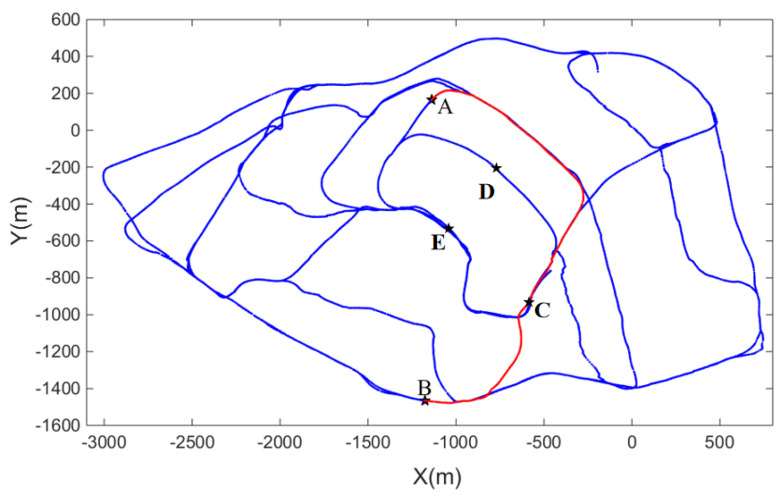
Path planning result of the third scene after the introduction of connectivity network.

**Table 1 biomimetics-08-00059-t001:** Path planning algorithm using episodic memory based on probabilistic selection.

1: **Input:** episodic cognitive map, sequence of episodic events, current episodic event *ε_cur_*, target episodic event *ε_tar_*. 2: **Output:** Sequence of episodic events {*ε_out_*} 3: **while** (*k* < *k_0_*) 4: **while** (*ε_cur_* ! = *ε_tar_*) 5: *n*= find the number of max connection-weight with *ε_cur_* 6: **for** *q* from 1 to *n* 7: compute distance of *e_q_* and *ε_tar_* 8: compute *P_q_* as (9) 9: **end for** 10: choose *ε_q_* as the next *ε* 11: compute distance between *ε_q_* and *ε_cur_* 12: make connection weights of *ε_q_* and *ε_cur_* as 0 13: *ε_cur_ *= *ε_q_* 14: **end while** 15: save the current sequence of episodic events 16: *k*++ 17: **end while** 18: choose the sequence of events with min of sum distance as {*ε_out_*}

**Table 2 biomimetics-08-00059-t002:** Comparisons of path planning algorithms.

	Without Probabilistic Model	Using Probabilistic Model
Length of the path (m)	169.7	70.8
Number of turns	5	4

**Table 3 biomimetics-08-00059-t003:** Comparisons of path planning.

Experimental Scenario	First Scene		Second Scene	
	Length of the Path (m)	Number of Turns	Length of the Path (m)	Number of Turns
Method of [39]	81.4	4	223.0	6
Our method	68.5	2	97.9	2

## Data Availability

Not applicable.

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
