# Peer review of "Bionic Path Planning Fusing Episodic Memory Based on RatSLAM"

_biomimetics, 2023, doi:10.3390/biomimetics8010059_

Round 1

Reviewer 1 Report

The article deals with current subject related to path planning of mobile robots. A bionic path planning algorithm based on RatSLAM was used to solve this kind of problem. Additionally, a neural network fusing historic episodic memory was used to improve the connectivity of the episodic cognitive map. In order to verify the developed solution, experimental tests were carried out.

The article requires major revision in the following scope.

Substantive remarks:

1.      The path planning of mobile robots can be divided into global planning or local planning (also called local navigation). Each of these types of planning involves different methods. Therefore, it should be made clear in the introduction what kind of planning is meant and addressed in the state of the art review.

2.      The review of the state of the art should be supplemented, additionally referring to the connectivity network model.

3.      Sections “2.2. Path planning using episodic cognitive map” and “3.1. The architecture of connectivity-improved episodic cognitive mapping” should be expanded as far as possible. In particular, all the methods should be formally described.

4.      In the description of formula (6) it is written that σx and σy represent the variance. Rather, it should be written “standard deviation”. Variance is the squared standard deviation.

5.      The developed solution was verified in a fairly trivial environment. Did the authors also conduct simulation or experimental studies in a more complex environment? If so, it would be appropriate to add such an example.

6.      The figures show paths without obstacles in the robot's environment. This makes it significantly more difficult to assess the effectiveness of path planning. At least an approximate arrangement of obstacles in the robot's surroundings should be added.

7.      The question arises why the method from [42] was used for the comparison? More reliable could be the choice of, for example, the A* algorithm or its modification, which guarantees the minimum path length.

8.      It is not clear from Figure 13 that the selected red path A-B is the shortest. Please verify and expand the justification.

9.      Section “5. Conclusion” is mainly a summary of the article. Within this section, the most important conclusions resulting from the conducted research should be described.

Editorial remarks:

1.      Spaces should be added before citing references, e.g. "performance [8-10]" should be written, not "performance[8-10]".

2.      The formatting of the M matrix should be corrected in dependence (1).

Reviewer 2 Report

The subject is interesting and is aligned with the readership and the themes of this journal. However, the paper does not include enough evidence to support the claim. The following bullet points include some suggestions to improve the manuscript to be publishable.

·       The paper must identify the gap that the research, which the authors conducted, fills.

·       Related work is too lengthy, but the more crucial issue is that the section does not have good organization. 

·       For readers to quickly catch your contribution, it would be better to highlight major difficulties and challenges, and your original achievements to overcome them, in a clearer way in abstract and introduction.

·       What are the other feasible alternatives? What are the advantages of adopting this technique over others in this case? How will this affect the results? More details should be furnished.

·       Some assumptions are stated in various sections. Justifications should be provided on these assumptions. Evaluation on how they will affect the results should be made.
 The discussion section in the present form is relatively weak and should be strengthened with more details and justifications.

·       There are some occasional grammatical problems within the text. It may need the attention of someone fluent in English language to enhance the readability.

·       I advise you just cite the papers in the literature review.

1.     Comparative Analysis of Low Discrepancy Sequence-Based Initialization Approaches Using Population-Based Algorithms for Solving the Global Optimization Problems

2.     A modified bat algorithm with torus walk for solving global optimization problems

·        Firstly, for section 1, authors should provide more specific comments of the cited papers after introducing each relevant work. What readers require is, by convinced literature review, to understand the clear thinking/consideration why the proposed approach can reach more convinced results. This is the very contribution from authors. In addition, authors also should provide more sufficient critical literature review to indicate the drawbacks of existed approaches, then, well define the main stream of research direction, how did those previous studies perform? Employ which methodologies? Which problem still requires to be solved? Why is the proposed approach suitable to be used to solve the critical problem? We need more convinced literature reviews to indicate clearly the state-of-the-art development.

Round 2

Reviewer 1 Report

The authors in the revised version of the manuscript took into account all recommendations, therefore the article is suitable for publication.

Reviewer 2 Report

The author has modified the changes in manuscript as pointed out earlier. I believe the paper overall is acceptable and will be of value to the research community. Now this paper can be published.